# In Vitro Angiogenesis Inhibition and Endothelial Cell Growth and Morphology

**DOI:** 10.3390/ijms23084277

**Published:** 2022-04-12

**Authors:** Arlinda Ljoki, Tanzila Aslam, Tina Friis, Ragnhild G. Ohm, Gunnar Houen

**Affiliations:** 1Department of Autoimmunology, Statens Serum Institut, DK-2300 Copenhagen, Denmark; arlinda_ljoki@hotmail.com (A.L.); tanzila14@hotmail.com (T.A.); tfs@ssi.dk (T.F.); 2Department of Drug Design and Pharmacology, University of Copenhagen, DK-2200 Copenhagen, Denmark; ragnhild.ohm@gmail.com; 3Department of Neurology, Rigshospitalet, DK-2600 Glostrup, Denmark

**Keywords:** angiogenesis, co-culture, HUVEC, inhibitor, morphology, NHDF, VEGF

## Abstract

A co-culture assay with human umbilical vein endothelial cells (HUVECs) and normal human dermal fibroblasts (NHDFs) was used to study whether selected angiogenesis inhibitors were able to inhibit differentiation and network formation of HUVECs in vitro. The effect of the inhibitors was determined by the morphology and the calculated percentage area covered by HUVECs. Neutralizing VEGF with avastin and polyclonal goat anti-VEGF antibody and inhibiting VEGFR2 with sorafenib and vatalanib resulted in the formation of HUVEC clusters of variable sizes as a result of inhibited EC differentiation. Furthermore, numerous inhibitors of the VEGF signaling pathways were tested for their effect on the growth and differentiation of HUVECs. The effects of these inhibitors did not reveal a cluster morphology, either individually or when combined to block VEGFR2 downstream pathways. Only the addition of *N*-methyl-*p*-bromolevamisole revealed a similar morphology as when targeting VEGF and VEGFR2, meaning it may have an inhibitory influence directly on VEGFR signaling. Additionally, several nuclear receptor ligands and miscellaneous compounds that might affect EC growth and differentiation were tested, but only dexamethasone gave rise to cluster formation similarly to VEGF-neutralizing compounds. These results point to a link between angiogenesis, HUVEC differentiation and glucocorticoid receptor activation.

## 1. Introduction

Cancer’s incidence has increased over many years and was estimated at approximately 19 million new cases worldwide in 2020, where cancer was responsible for 8.2 million deaths with about 90% being caused by metastases [1,2,3]. Tumor growth and metastases are major factors associated with a poor prognosis; these depend on angiogenesis, which is one of several hallmarks of cancer [4,5,6,7,8]. An effective approach to prevent tumor progression is, therefore, to inhibit angiogenesis, which is stimulated by a variety of factors, the main trigger being a low oxygen concentration (hypoxia) [4,5,6,7].

The overall process of angiogenesis includes extracellular matrix degradation/remodeling, formation of capillaries by “sprouting” of existing capillaries and finally, consolidation of larger vessels by smooth muscle cells [4,5,6,7]. An influx and differentiation of endothelial precursor cells (EPCs) may also contribute to the buildup of a tumor vessel supply, denoted vasculogenesis, which may occur concomitantly and synergistically with angiogenesis [7,9]. Angiogenesis/vasculogenesis requires the action of multiple growth factors and cytokines, which activate intracellular signaling pathways, synthesis of proteases, inhibitors, and other molecules, all controlled by an array of transcription factors [4,5,6,7,10,11,12,13,14,15,16,17]. A key molecule in this is vascular endothelial growth factor (VEGF), which is synthesized by hypoxic cells and binds to VEGF receptor 2 (VEGFR2) on endothelial cells, activating its intrinsic kinase activities. This, in turn, activates several downstream signaling pathways, which regulate several transcription factors (TFs) supporting the survival, proliferation, migration and differentiation of endothelial cells (ECs) [10,11,12]. Thus, the downstream signaling pathways play an essential role in the regulation of multiple genes important for endothelial survival, proliferation, migration and differentiation [12,14,15,17].

The first anti-angiogenesis drug, approved in 2004 by the US Food and Drug Administration (FDA), was bevacizumab (avastin), a recombinant humanized clinically effective neutralizing monoclonal antibody (MAb) against VEGF for first- or second-line treatment of metastatic colon cancer [18,19]. Avastin may induce side effects such as hypertension, thromboembolic events and gastrointestinal bleeding; therefore, a search for low-molecular-weight drugs with limited toxicity and side effects has been a major goal in angiogenesis research [18,19].

Clinically approved low-molecular-weight angiogenesis inhibitors include the tyrosine kinase inhibitors sorafenib, vatalanib and others [20,21]. Sorafenib is a receptor tyrosine kinase inhibitor (RTKI), which was originally developed as an inhibitor of Raf-1 and was later shown to inhibit multiple other kinases and to have a high affinity with VEGFR2 [22,23,24]. Vatalanib is an orally active multi-target inhibitor with a high VEGFR2-binding affinity [20,23,25,26]. Side effects of these tyrosine kinase inhibitors are relatively few, but may still be serious, including fatigue, nausea, anemia, thrombocytopenia and others, and resistance may eventually develop [20,21,22,23,24,25,26,27]. Multiple other low-molecular-weight compounds may exhibit anti-angiogenic activity in various assays but may not be clinically effective and often show a prohibitive side-effect profile. Therefore, in vitro assays that can identify compounds with potential anti-angiogenic effects similar to VEGF-blocking agents and with limited general toxicity are of great interest. Here, a co-culture angiogenesis assay with human umbilical vein endothelial cells (HUVECs) and normal human dermal fibroblasts (NHDFs) [7,28,29], which fulfills the requirements described above, was used to investigate the effect of several known and potential angiogenesis inhibitors on HUVEC growth and morphology.

## 2. Results

### 2.1. Angiogenesis Inhibitors’ Effects on Endothelial Cell Morphology and Network Formation

To investigate the influence of angiogenesis inhibitors on HUVEC morphology in vitro, a wide range of inhibitors were chosen according to the different VEGF-induced signaling pathways. Furthermore, nuclear receptor ligands and miscellaneous compounds were tested to reveal potential effects of other signaling pathways, and some inhibitors were tested on both HUVECs and GFP-HUVECs (Table 1).

The co-culture angiogenesis assay revealed the inhibitors’ effects on HUVEC morphology using CD31 MAb staining and allowed a quantitative measurement by image analysis of the percentage of area covered by HUVECs. Figure 1A,B shows relevant controls including DMSO 0.1%, TBS and co-culture EBM-2 medium, reflecting how the various inhibitors were dissolved in DMSO and diluted with TBS before testing, and Figure 1C illustrates the effect of omitting VEGF from the assay. While there were no statistically significant differences between the controls, all showing extensive network formation, omission of VEGF abrogated network formation and resulted in the HUVECs forming isolated clusters and cords.

Figure 2 illustrates the growth and morphology of HUVECs in the co-culture angiogenesis assay in the presence of selected inhibitors, and Table 1 summarizes the effects of all inhibitors tested. The representative images in Figure 2 were chosen by the lowest concentration that gives a clear visual effect on the morphology without showing general toxicity. Figure 2A shows the inhibitors’ effect on the morphology, whereas Figure 2B reveals the percentage of area covered by HUVECs as detected by CD31 MAb staining image analysis. In general, HUVECs and GFP-HUVECs yielded the same results, although GFP-HUVECs were slightly more sensitive to the inhibitors (Table 1). For every repeated experiment with an inhibitor, the HUVECs revealed the same morphology but the area coverage of cells differed somewhat between experiments. Table 1 includes a grading, based on the percentage of area covered by (GFP-)HUVECs. To evaluate the effect of the inhibitors, it was important to compare both the percentage of area covered by the HUVECs and their morphology. The cells that formed clusters could still cover a large area, if proliferation was not affected, illustrating the importance of comparing the quantitative image analysis results with the qualitative results, or the morphology.

### 2.2. VEGF and VEGFR Inhibitors

The coverage of HUVECs after treatment with different concentrations of bevacizumab (avastin)—2.5 mg/mL, 250 µg/mL and 25 µg/mL—was approximately 5%, 8% and 6% (mean of four experiments), respectively, compared to the control, which showed 16% coverage by HUVECs. With a high concentration (2.5 mg/mL), the HUVECs formed clusters, but with a lower concentration (25 µg/mL), the cells were able to differentiate and proliferate and form cords. Sorafenib and vatalanib at a concentration of 1 µM gave rise to cluster formation and resulted in approximately 3% coverage of HUVECs (Figure 2, Table 1). Identical results were seen with GFP-HUVECs (Table 1).

### 2.3. Levamisole and Its Derivatives

Levamisole has previously shown inhibition of angiogenesis in vitro, when HUVECs were propagated on NHDFs, and inhibition of tumor growth in vivo [30,31]. Levamisole and various derivatives were tested at 1 mM and 100 µM, respectively, which were the highest concentrations with an inhibitory effect without being lethal (Figure 3). Levamisole and *N*-methyl-levamisole induced clusters and cords; *N*-methyl-*p*-bromolevamisole gave rise to only oval clusters, whereas *p*-bromolevamisole was the least effective. Similar results were seen for GFP-HUVECs (Table 1).

### 2.4. VEGFR Downstream Inhibitors

The calcineurin inhibitors cyclosporin and mycophenolate mofetil were either ineffective or had a weak inhibitory effect, revealing short cords at the highest concentration. The MAPK inhibitor SB203580 and the MEK inhibitor PB98059 showed a dose-dependent inhibition of network formation, while the MEK inhibitor trametinib gave short cords (HUVECs) or had a cytotoxic effect (GFP-HUVECs). The RAF inhibitor vemurafenib was ineffective or cytotoxic at the highest concentration. Regarding the PI3K-mTOR-AKT pathway, LY333531 (10 µM), gedatolisib (300 nM), PI-103 (1 µM) and NSC 87877 (100 µM) abrogated network formation and induced a short cord morphology with different percentages of area. Only PI-103 at a concentration of 10 µM resulted in dot morphology, indicating that the cells were not able to differentiate or proliferate (Table 1, Figure 2).

### 2.5. β-Catenin Inhibitors

Experiments replacing the NHDF layer with extracellular matrix components showed that the HUVECs needed cell-cell interaction with live NHDFs for network formation, as also observed previously [28]. Since cadherins might be involved in this and since β-catenin might play an important role by interacting with the cytoplasmic part of the cadherins, we tested four β-catenin inhibitors, IWR-1, JW74, PNU74654 and ICRT3, at a concentration of 10 µM. The experiment was carried out twice, and the area covered by the HUVECs was approximately equal to the control (Table 1, Figure 4). Thus, the β-catenin inhibitors had no significant inhibitory effect on the HUVECs at 10 µM, which was the highest obtainable concentration. Identical results were obtained for GFP-HUVECs (Table 1).

### 2.6. Combinations of Angiogenesis Inhibitors

When inhibiting VEGFR2 with sorafenib and vatalanib, the HUVECs revealed a non-differentiated cluster morphology and no network formation (Table 1, Figure 3). To study if the same effect could be achieved through inhibition of downstream pathways, several pathways involved in VEGFR2 signaling were targeted simultaneously with combinations of different inhibitors (Table 2). The chosen concentrations of the combined inhibitors were based on the inhibitors individually revealing non-toxicity and at the same time showing HUVEC inhibition. Since some inhibitors with the same target gave rise to different HUVEC morphologies, some of them were tested several times in combinations, e.g., LY294002 and gedatolisib, which both inhibit PI3K. However, none of the combinations resulted in cluster formation; instead, they created more like a dot morphology, cords or an inhibited network, and when combining the inhibitors, the strongest inhibitor appeared to determine the morphology. The morphology obtained for each inhibitor individually is described in Table 1 and some are shown in Figure 2B.

### 2.7. Nuclear Receptor Ligands and Miscellaneous Compounds

The results indicated that VEGF is a major regulator inducing a capillary-like network. The finding that none of the combined inhibitors showed cluster formation might indicate that there are other uninvestigated pathways affected by VEGF. Therefore, we tested several compounds that might affect HUVEC growth and differentiation. When the HUVECs were treated with suramin (100 µM), they formed short cords, but when tested in a lower concentration (10 µM), they started to form a network (Table 1, Figure 2). Retinol, retinoic acid and anacardic acid did not show any inhibitory effect, as the ECs formed a network, when compared to butyric acid, valproic acid and calcitrol (25-hydroxyvitamin D), with only in a high concentration giving rise to cords without branches (Table 1). Surprisingly, HUVECs treated with dexamethasone showed clusters as determined by the morphology and the calculated percentage coverage of area (Figure 5). The morphology of HUVECs was clearly affected by every concentration, for which the cells appeared as clusters compared to the control. The experiment revealed that even 1 pM was efficacious at inhibiting the differentiation of HUVECs, which formed clusters and cords; however, there was not necessarily the same degree of clusters as at the highest concentrations (1–100 µM).

## 3. Discussion

In this work, we have investigated the effect of VEGF and different inhibitors on HUVECs in co-culture with NHDFs. An advantage of the assay used is that it allows both a qualitative (morphology) and a quantitative evaluation of the effect of different compounds [28,29]. When propagated on NHDFs in the presence of VEGF, the HUVECs differentiated and formed a capillary-like network, whereas in the absence of VEGF or in the presence of neutralizing monoclonal (avastin/bevacizumab) or polyclonal VEGF antibodies, the HUVECs formed clusters of less differentiated cells (Figure 1 and Figure 2). Thus, in some respects, VEGF may be viewed as a differentiation factor as well as a growth factor.

Activation of VEGFR2 is known to involve autophosphorylation and phosphorylation of several targets, which initiate downstream signaling pathways [10,11,12,13,14,15,16,17,32,33,34,35]. Figure 6 provides an overview of VEGFR2 signaling pathways and the effects of the compounds tested here on HUVEC morphology and/or growth.

### 3.1. Inhibition of VEGF and VEGFR Signaling

The cluster morphology, as seen when VEGF was omitted or neutralized with avastin or polyclonal VEGF antibodies, was observed with the tyrosine kinase inhibitors sorafenib and vatalanib but also with *N*-methyl-*p*-bromolevamisole. Sorafenib has been shown to block VEGFR2, PDGFR-β and the MAPK pathway [22,23,24,36,37,38], whereas vatalanib inhibits all VEGFRs but with a higher potency toward VEGFR2 [20,23,25,26,39,40]. The higher potency toward VEGFR2 was reflected in that vatalanib was more effective at inducing clusters in lower concentrations than sorafenib. Wood et al. [41] showed that the formation of capillary-like sprouts from pieces of rat aorta was decreased in a dose-dependent manner up to 1 μM, which did not have cytotoxic or antiproliferative effects on cells that did not express VEGFR. This is compatible with our results.

The effect of levamisole and its derivatives on angiogenesis is still unclear, but the results obtained here indicate that they may affect VEGFR2. Since levamisole is clinically approved and its toxicology profile is known, development of more potent derivatives revealing cluster formation is of great interest. Hansen et al. [30] showed that *p*-bromolevamisole and *N*-methyllevamisole were more efficient inhibitors than levamisole. Our results showed that *p*-bromolevamisole and levamisole gave rise to the same HUVEC morphology, but the area percentage differed (10.4% and 8.4%, respectively). Furthermore, *N*-methyllevamisole (7.4%) and *N*-methyl-*p*-bromolevamisole (6.4%), proved to yield efficacious morphological inhibition of HUVEC differentiation by forming oval clusters.

Inhibitors of VEGFR2 downstream signaling pathways were less effective than the VEGFR2 inhibitors. Inhibition of SHP2 by NSC 87877 at the highest concentration (100 µM) showed a short cord morphology, with a differentiated appearance but without cell-cell interactions. PTP IV had a lethal effect on HUVECs and NHDFs at concentrations of 100 µM and 10 µM, but at 1 µM, the HUVECs formed long cords, whereas with lower concentrations, more network formation was seen (Table 1). This is in contrast to Sylvest et al. [42], who reported a cluster morphology when HUVECs were treated with PTP IV at 10 µM. However, the morphology observed by Sylvest et al. was actually somewhat different when compared to levamisole. These two phosphatase inhibitors may affect a wide range of different phosphatases [43], which could be a reason for somewhat different morphological results in different studies.

Activation of the MAPK signaling pathway may increase the levels of HIF-1α [44]. LY333531, trametinib, vemurafenib, SB203580 and PD98059 have been reported to inhibit this cascade and most had an inhibitory effect in the co-culture assay, inducing either a short cord morphology (LY333531, trametinib, mycophenolate mofetil (Figure 2, Table 1)) or inhibited network formation (SB203580, PD98059). Vemurafenib had no effect in the assay, presumably because this inhibitor, according to Spagnolo et al. [45], only works in cancer cells with a V600E B-RAF mutation.

The PI3K-AKT pathway is activated in the majority of human cancers and is involved in the regulation of angiogenesis [46,47]. The inhibitors XL147, PI-103, LY204002, ZSTK474, wortmannin and gedatolisib target the PI3K pathway and most had an effect in the co-culture assay. At a concentration of 10 µM, PI-103 and ZSTK474 revealed a dot morphology (Figure 3, Table 1). PI-103 and ZSTK474 have also been shown to inhibit mTOR, which makes them less specific for PI3K. HUVECs treated with gedatolisib and XL147 gave a mixture of dots and short cords, whereas LY294002 gave rise only to cords. Wortmannin did not affect the HUVECs and formed a network.

Kong et al. [48] identified ZSTK474 as a PI3K inhibitor, which had an inhibitory effect in vivo against a human cancer xenograft without observable toxicity and was at least 30-fold more effective than LY294002. In this project, LY294002 and ZSTK474 at a concentration of 100 µM were toxic to both HUVECs and NHDFs. At 10 µM, ZSTK474 was more effective than LY294002, because the HUVECs appeared as dots, in contrast to cords, when treated with 10 µM LY294002. This observation was similar to Kong et al. [48], who found ZSTK474 to be more effective as an angiogenesis inhibitor. Perifosine, which inhibits AKT and prevents translocation to the cell membrane, showed an inhibitory effect on network formation [49].

### 3.2. Miscellaneous Inhibitors

Suramin has been shown to inhibit angiogenesis by blocking specific receptor binding of PDGF, bFGF, VEGF and neuropilin [50,51,52] and might have been expected to show a cluster morphology. However, we found that 10 µM and 50 µM of suramin showed an inhibitory effect on cellular network formation, by forming cords with interactions, and a complete absence of network at 100 µM, by forming only cords (Table 1, Figure 2). These findings were compatible with other published studies; Friis et al. [7,28] found that 100 µM of suramin was an effective inhibitor of HUVEC proliferation, which gave rise to a morphology of EC as short cords, and Prigozhina et al. [53] observed an inhibitory effect of suramin at a concentration of 8 µM and higher on HUVECs when cultured on matrigels. Furthermore, Gagliardi et al. reported that suramin inhibited tumor growth by inhibition of angiogenesis [51].

Since many tumors may also produce other angiogenic factors, blocking VEGF may not be efficacious to totally inhibit tumor angiogenesis, and some types of cancers may switch to another angiogenic factor by adaptive mutations [54]. Therefore, we investigated inhibitors blocking other signaling pathways, including the JAK/STAT signaling pathway, which has been shown to regulate cell growth, proliferation, differentiation and apoptosis, and that is important in the signal transduction of growth factors [55,56]. However, the inhibitors WP1066, Ruxolitinib, AG-490, SH-4-54 and NSC 74859 were not able to induce cluster formation of the HUVECs or inhibit network formation (Table 1).

We further investigated the potential and mechanism of some Wnt/β-catenin inhibitors, IWR1, JW74, ICRT3 and PNU-74654 (Figure 4). The central player in the canonical Wnt cascade is β-catenin, and in epithelia, it is a component of adherens junctions [57]. Previous studies have shown that inhibiting β-catenin induces an anti-angiogenic effect by affecting tumor formation and growth [58,59]. The results here showed that inhibition of β-catenin did not have a clear effect on HUVECs’ morphology, which may indicate that β-catenin is not the only adhesion junction that must be inhibited to get a full inhibition of the interaction between HUVECs and NHDFs.

### 3.3. Combinations of Inhibitors

Combinations of the inhibitors were tested with the intention of achieving complete inhibition of the downstream pathways involved in VEGFR2 signaling. Inhibition of network formation was achieved by several combinations. However, none of the combined inhibitors were efficacious with respect to induction of clusters, not even when all the known pathways were blocked (Table 2). As mentioned earlier, clusters were only induced when one of the inhibitors in a combination was either anti-VEGF, VEGFR2 inhibitor or when VEGF was omitted from the media. The VEGFR2 signaling pathway involves several kinases and phosphatases; therefore, the reason for not achieving clusters may be that the inhibitors are not effective enough or that there might be yet another pathway affecting HUVEC differentiation.

### 3.4. Vitamin D and Dexamethasone

25-hydroxyvitamin D3 monohydrate (calcitriol) and dexamethasone have similar structures and have also shown anti-tumor activity. Calcitriol inhibits proliferation, promotes cell differentiation, regulates cell cycle arrest, induces apoptosis and has an anti-inflammatory effect, which inhibits cell growth in various types of cancer cells [60,61,62]. When calcitriol binds to the vitamin D receptor (VDR), it forms a complex, which induces or inhibits gene transcription by affecting transcription factors [63,64]. Our study has shown that HUVECs treated with 1 mM calcitriol revealed cords, which indeed indicates differentiated non-proliferating cells. In lower concentrations of calcitriol, the HUVECs formed a network even with the addition of vitamin D-binding protein (DBP), which transports and thereby helps calcitriol to its target (Table 1).

Several studies have demonstrated that dexamethasone may induce inhibition of angiogenesis through the intracellular glucocorticoid receptor (GR), which is a member of the superfamily of nuclear receptors [65,66,67,68,69]. The GR is found in the cytoplasm, and when a ligand, e.g., dexamethasone, binds, it is transported into the nucleus where it regulates gene transcription, by inhibiting the expression of the nuclear transcription factors, AP-1 and NF-κB [70]. Jia et al. [56] demonstrated that AP-1 is activated by VEGF. AP-1 is known to be critical for cell growth, differentiation and cell-cell interaction, and by inhibiting AP-1, it may be effective to treat cancer via inhibition of VEGF signaling. Our study demonstrated that HUVECs treated with dexamethasone clearly induced clusters in various sizes (Figure 4). Although the mechanism by which this occurs is still poorly understood, we assume that it involves changes in the activity of several transcription factors regulated by the GR.

### 3.5. Weakness of the Assay

Table 1 shows that HUVECs treated with some of the inhibitors were clearly affected and formed either clusters, cords, dots or an inhibited network. The percentages of area covered by HUVECs were 3.0% and 3.2%, respectively, and with a grading ++ for both sorafenib and vatalanib at 1 µM (Figure 2A, Table 1). The treatment of HUVECs with suramin (100 µM) (3.2%) or sorafenib (1 µM) (3.0%) gave approximately the same percentage of area coverage but resulted in different morphologies. It is notable that the percentage of area covered by HUVECs and the grading do not indicate the HUVECs’ morphology and may result in a misleading evaluation of the inhibitor. A high percentage of area does not necessarily correlate with network formation, or a low percentage of area does not necessarily reflect cluster formation. Therefore, it is important to compare the quantitative image analysis results with the morphology.

GFP-HUVECs were investigated to continuously monitor the progress of the assay. However, the GFP-HUVECs appeared to be more sensitive to several compounds, presumably due to the presence of the GFP-expressing plasmid, thus yielding erroneous results for several compounds.

## 4. Materials and Methods

### 4.1. Materials

MilliQ water Equipment, Purelab Ultra was sourced from Krüger ELGA LabWater (Glostrup, Denmark). Dimethylsulfoxide (DMSO) (>99.99%) was sourced from WAK-Chemie (Stainbach/Ts, Germany). Ethanol (96%) was sourced from CCS Healthcare AB (Borläng, Sweden). Phosphate-buffered saline (PBS, 10 mM NaH_2_PO_4_•H_2_O/NaH_2_PO_4_•2H_2_O, pH 7.4, 0.15 M NaCl), tris-buffered saline (TBS, 50 mM tris, pH 7.4, 0.15 M NaCl) and tris-tween-NaCl buffer (TTN, 50 mM tris buffer, 1% tween 20, 0.3 M NaCl, pH 7.5) were sourced from Statens Serum Institut (Copenhagen, Denmark).

We sourced 5-bromo-4-chloro-3-indolyl phosphate/nitro blue tetrazolium tablets (BCIP/NBT), levamisole hydrochloride, *p*-bromolevamisole oxalate, AP-conjugated goat anti-mouse IgG (1 mg/mL), cyclosporin A, LY333531 hydrochloride, mycophenolate mofetil, suramin sodium salt, dexamethasone, butyric acid, retinol, retinoic acid, 25-hydroxyvitamin D3 monohydrate, valproic acid sodium salt and the β-catenin inhibitors IWR-1, JW74, ICRT3, PNU-74654 and nigrosine stain solution (0.2%) for microscopy from Sigma Aldrich (St. Louis, MO, USA).

Gedatolisib, vatalanib, sorafenib, vemurafenib, trametinib, perifosine, PI-103, XL147 and ZSTK474 were sourced from Selleckchem (Houston, TX, USA).

Wortmannin, SB203580, PD98059, PTP IV and STAT 5 were sourced from Calbiochem (Darmstadt, Germany). NSC 87877 and tautomycetin were sourced from Tocris Bioscience (Bristol, UK).

LY294002 was sourced from Cell Signaling Technology (Beverly, MA, USA).

Anacardic acid was sourced from Santa Cruz Biotechnology (Heidelberg, Germany).

Avastin (bevacizumab) was sourced from Roche (Grenzach-Wyhlen, Germany). WP1066, ruxolitinib, AG-490, SH4-54 and NSC 74859 were sourced from MedChem Express (Monmouth Junction, NJ, USA).

Human umbilical vein endothelial cells (HUVECs) and normal human dermal fibroblasts (NHDFs) were sourced from Lonza (Walkersville, MD, USA). Green fluorescent protein-expressing HUVECs (GFP-HUVECs) were sourced from Angio-Proteomie (Boston, MA, USA).

Fetal bovine serum (FBS) was sourced from Sera Scandia (Hellerup, Denmark).

Murine monoclonal anti-cluster of differentiation 31 (CD31) antibody 0.2 mg/mL was sourced from Monosan (Uden, The Netherlands).

Nunclon EASY flasks with a filter lid (T25 cm^2^ and T80 cm^2^) and 96-microwell Nunclon polystyrene plates were sourced from Nunc (Roskilde, Denmark).

Fibroblast growth medium-2 (FGM-2) containing fibroblast basic medium (FBM), recombinant human fibroblast growth factor-basic (rhFGF-B), insulin, gentamicin sulfate/amphotericin (GA-1000), FBS and endothelial cell growth medium (EGM-2) containing endothelial cell basic medium-2 (EBM-2), recombinant human epidermal growth factor (rhEGF), hydrocortisone, GA-1000, FBS, recombinant human vascular endothelial growth factor (rhVEGF), rhFGF-B, recombinant-3 insulin-like growth factor 1 (R^3^-IGF-1), ascorbic acid and heparin, 4-(2-hydroxyethyl) piperazine-1-ethanesulfonic acid (HEPES)-buffered saline solution (BSS), trypsin, ethylene diamine tetraacetic acid (EDTA) and trypsin-neutralizing solution (TNS) were sourced from Lonza (Walkersville, MD, USA). rhFGF-B (10 µg/mL), rhVEGF (100 µg/mL) and purified neutralizing goat anti-hVEGF antibodies (IgG, 400 µg/mL) (polyclonal goat anti-human VEGF antibody) were sourced from R&D Systems (Minneapolis, MN, USA). The following media were made from FGM and EGM-2: NHDF standard medium containing insulin, rhFGF-B, FBS and GA-1000 in FBM. HUVEC standard medium containing ascorbic acid, heparin, R^3^-IGF-1, rhEGF, rhVEGF, rhFGF-B, hydrocortisone, FBS and GA-1000 in EBM-2 medium. TFSM2 medium (used with extracellular matrix components) containing 100 µL ascorbic acid, 100 µL heparin, 100 µL GA-1000, 2.0 mL FBS, 100 µL rhEGF, 10 µL rhFGF-B (10 µg/mL) and 10 µL rhVEGF (100 µg/mL) in 100 mL EBM-2 medium. TFSM2^+10%^ medium used for co-culture containing 110 µL ascorbic acid, 110 µL heparin, 110 µL GA-1000, 2.2 mL FBS, 110 µL rhEGF, 11 µL rhFGF-B (10 µg/mL) and 11 µL rhVEGF (100 µg/mL) in 100 mL EBM-2 medium.

### 4.2. Chemical Syntheses

*N*-methyl-levamisole was synthesized essentially as described previously [30]. Levamisole HCl (50 mg, 0.21 µmol) was dissolved in aqueous NaOH (2 M, 25 mL) and the aqueous phase was extracted with diethyl ether (3 × 25 mL). The combined organic layers were dried over Na_2_SO_4_, filtered and the solvent was removed under reduced pressure. The residue was taken up in THF (dry, 2 mL) in a nitrogen atmosphere, iodomethane (20 µL, 8.8 µmol) was added and the mixture was stirred under reflux for 23 h. The reaction was then cooled to room temperature and water (15 mL) was added. The aqueous layer was washed with CH_2_Cl_2_ (3 × 15 mL) and then lyophilized. The crude material was purified by preparative HPLC (as described below) to give the *N*-methyllevamisole TFA salt (28 mg, 0.084 µmol, 40%) as a white fluffy material. ^1^H NMR (400 MHz, DMSO-d6) δ 7.66–7.50 (m, 5H), 5.65 (dd, *J* = 10.4, 9.0 Hz, 1H), 4.33 (t, *J* = 10.4 Hz, 1H), 4.18–4.10 (m, 2H), 4.00–3.84 (m, 2H), 3.78 (dd, *J* = 10.4, 9.0 Hz, 1H), 2.92 (s, 3H). MS (ESI) *m/z*: calc for C_12_H_15_N_2_S^+^ [M^+^] 219.1, found 219.0.

Para-bromo-*N*-methyl-levamisole was synthesized following a procedure similar to the one described above. (–)-*p*-Bromolevamisole oxalate (49 mg, 0.13 µmol) and K_2_CO_3_ (20 mg, 0.15 µmol) were mixed in a flame-dried flask under a nitrogen atmosphere. THF was added (dry, 2 mL), followed by iodomethane (10 µL, 4.4 µmol) and the reaction was stirred under reflux for 24 h. More THF (dry, 1 mL) was then added, followed by another portion of K_2_CO_3_ (20 mg, 0.15 µmol) and iodomethane (10 µL, 4.4 µmol). The reaction was heated to reflux for another 5 h. The reaction was then cooled to room temperature and filtered. The filter cake was washed with THF (10 mL) and water (10 mL). The solvent was removed under reduced pressure and the residue was purified by preparative HPLC (as described below) to give the *N*-methyl-*p*-bromolevamisole TFA salt (2 mg, 0.0049 µmol, 4%) as a white fluffy material. ^1^H NMR (400 MHz, DMSO-d6) δ 7.71 (d, *J* = 8.3 Hz, 2H), 7.50 (d, *J* = 8.3 Hz, 2H), 5.57 (t, *J* = 9.7 Hz, 1H), 4.23 (t, *J* = 10.4 Hz, 1H), 4.08–4.01 (m, 2H), 3.91–3.76 (m, 2H), 3.72–3.63 (m, 1H), 2.84 (s, 3H). MS (ESI) *m/z*: calc for C_12_H_14_N_2_SBr^+^ [M^+^] 297.0, found 297.1. 

Preparative reversed-phase HPLC separations were carried out on a Phenomenex Luna C18 column (5 mm) using an Agilent system, consisting of two preparative scale pumps, an autosampler and a multiple-wavelength UV detector. A gradient consisting of eluent A [H_2_O–MeCN–TFA (95:5:0.1)] and eluent B [H_2_O–MeCN–TFA (5:95:0.1)], rising linearly from 5% to 80% of B over 40 min, was used. The eluent flow rate was maintained at 20 mL/min and injection volumes were 0.9 mL.

For LC-MS analysis, an Ultimate 3000 UHPLC system (Dionex, Sunnyvale, CA, USA) was used. Separations were performed on a 62 mm × 3 mm Kinetex C18 column (Phenomenex, Torrance, CA, USA), using a linear water–acetonitrile gradient (both buffered with formic acid, 1%) at a flow of 0.4 mL min, starting from 10% acetonitrile and increasing to 100% over 10 min. MS was performed in ESI^+^ with a data acquisition range of 10 scans per sec at *m/z* 100–1000. The MS was calibrated using sodium formate automatically infused before each analytical run, providing a mass accuracy of less than 0.5 ppm in MS mode.

### 4.3. Methods

Cells were stored at −135 °C and thawed immediately in a 37 °C water bath. When freezing the cells at −135 °C, the amount was 3 × 10^5^ cells/vial in freezing medium composed of FBS with 5% DMSO. All procedures took place at room temperature unless otherwise indicated and were performed on HUVECs within the first three passages and for NHDFs within the first five passages.

#### 4.3.1. Cell Preparation

NHDFs were cultured in T-80 cm^2^ Nunc culture flasks with a filter in 10 mL NHDF standard medium (FGM-2). HUVECs were cultured in T-25 cm^2^ Nunc culture flasks with a filter in 10 mL HUVEC standard medium (EGM-2) in a BDD 6220 incubator (Heraeus, Hanau, Germany) at 37 °C, 5% CO_2_ and 90% humidity.

After 24 h of culturing, the culture medium was changed to new standard medium to remove the DMSO, and 2–3 days later, the culture medium was again replaced or the cells were trypsinized off and propagated. The cells were incubated until they were confluent (approximately 3–4 days).

Before harvesting the cells with T-EDTA, they were washed twice with 5 mL HEPES-BSS. Then, 2.5 mL T-EDTA was added and incubated at 37 °C to release them from the culture surface. The cells were inspected under a microscope after 4–5 min to make sure they were released, and 2.5 mL TNS was added to neutralize the T-EDTA. The suspension was transferred to a 50-mL tube and centrifuged for 10 min at 1000 rpm in a Megafuge 2.0 (Heraeus, Hanau, Germany). The NHDF pellet was resuspended in 1 mL NHDF standard medium and the HUVEC pellet was resuspended in 1 mL TFSM2^+10%^ medium. The cells were counted with a mixture of 25 µL nigrosine (0.2%) and 25 µL cell suspension using a hemocytometer (Profondeur, Neubauer, Germany) and an Olympus CX31 microscope (Tokyo, Japan). To achieve a concentration of 10^3^ cells per well, for both NHDFs and HUVECs, a calculated volume of either NHDFs or TFSM2^+10%^ medium was added.

#### 4.3.2. Co-Culture Assay

In each well of a 96-microwell plate, 10^3^ NDHFs were seeded at 37 °C, 5% CO_2_ and 90% humidity. After three days of culturing, they created a dense layer in the bottom of the well. The medium was removed and 35 µL TFSM2^+10%^ medium was added. (GFP-)HUVECs (10^3^) were seeded on top of the dense layer of NHDFs. Then, 15 µL of the different angiogenesis inhibitors was added to selected wells. EMB media, TBS and 0.1% DMSO (in TBS) were used as controls, since the inhibitors were dissolved in 99.99% DMSO and diluted in TBS. The co-culture was incubated for three days at 37 °C, 5% CO_2_ and 90% humidity.

The same procedure was performed with GFP-HUVECs and with HUVECs seeded on top of extracellular matrix components, but with TFSM2 medium.

After three days, the medium was removed from the wells and the plates were washed with 200 µL PBS per well, followed by 15 min fixation with 100 µL of −20 °C 96% ethanol. The wells were washed for 3 × 1 min with 200 µL TTN buffer and non-specific binding was blocked by incubation with TTN buffer for 1 h. The primary MAb against CD31 was diluted with TTN buffer (1:500) to 1 µg/mL and added to the wells for 1 h at room temperature or at 4 °C overnight. The cells were washed again 3 × 1 min with 200 µL TTN buffer, and the secondary antibody (alkaline phosphatase (AP)-conjugated goat anti-mouse IgG) (1 µg/mL, dilution 1:1000) was added to the wells and incubated for 1 h. Then, again, the wells were washed 3 × 1 min with TTN buffer. Bound antibody was visualized by incubation for 1 h with 100 µL staining solution (1% BCIP/NBT with 1 mM levamisole in milliQ water) in each well. After staining, the wells were washed three times with 200 µL milliQ water. The BCIP/NBT staining was visualized using a model IX70 microscope, while GFP-HUVECs were inspected by fluorescence microscopy with a model U-RFL-T microscope (4× magnification). Then, data were analyzed with the Cell D computer imaging program (Olympus, Tokyo, Japan).

Table 3 shows the grading used for evaluating the effects of inhibitors. The morphology of HUVECs that form a capillary-like structure with long cords and crossing branches with interactions is called a network. HUVECs that form round and oval cells in variable sizes refer to clusters, which indicates non-differentiated cells. HUVECs that form strings without a network are called cords. Short cords are defined as single cells just starting to elongate/proliferate, while long cords are defined as longer cords not forming a network. Dots refer to small, round, single cells, which are not able to grow. Differentiation refers to clear changes in the morphology, which result from changes in gene expression. Proliferation refers to multiplication of the cells, which means an increased cell amount. Elongation refers to an increased cell volume.

General cytotoxicity was evaluated by visually (via microscopy) inspecting the integrity of the NHDF layer on which the HUVECs were cultured [28,29].

#### 4.3.3. Statistical Analysis

Statistical analyses were carried out using R, version 4.1.3 (R Foundation, www.r-project.org, accessed on 4 April 2022). Normality was checked using the Shapiro–Wilks test and Bartlett’s test for homogeneity of variance. In the case of heteroscedasticity, data were log-transformed. One-way analysis of variance (ANOVA) was used, or in the case of two or three observations, the Kruskal–Wallis test was used instead. In the case of significance (usually *p* < 0.05), adjustments for multiple comparisons were done using Dunnett’s test comparing several treatments with the control, along with the Conover–Iman test of multiple comparisons using rank sums. Exact *p*-values and 95% confidence intervals were reported, except for very low values.

## 5. Conclusions

Despite the efficacious performance of VEGF-targeted therapy, which has become an important treatment option, unfortunately, some patients exhibit resistance and do not respond to the therapy [23]. Therefore, the results described here may be important in the search for clinically effective inhibitors of angiogenesis and in furthering our understanding of VEGF cellular actions. Since the formation of clusters is observed only when neutralizing VEGF, when inhibiting VEGFR2 and via dexamethasone treatment, but not in the VEGF downstream pathways, there could be a link to an uninvestigated pathway, which influences differentiation. Dexamethasone demonstrated a clear morphological anti-angiogenic effect in this in vitro co-culture angiogenesis assay; therefore, it may be a candidate for further clinical trials. Further studies are needed to elucidate the inhibitory mechanism of dexamethasone to inhibit ECs.

## Figures and Tables

**Figure 1 ijms-23-04277-f001:**
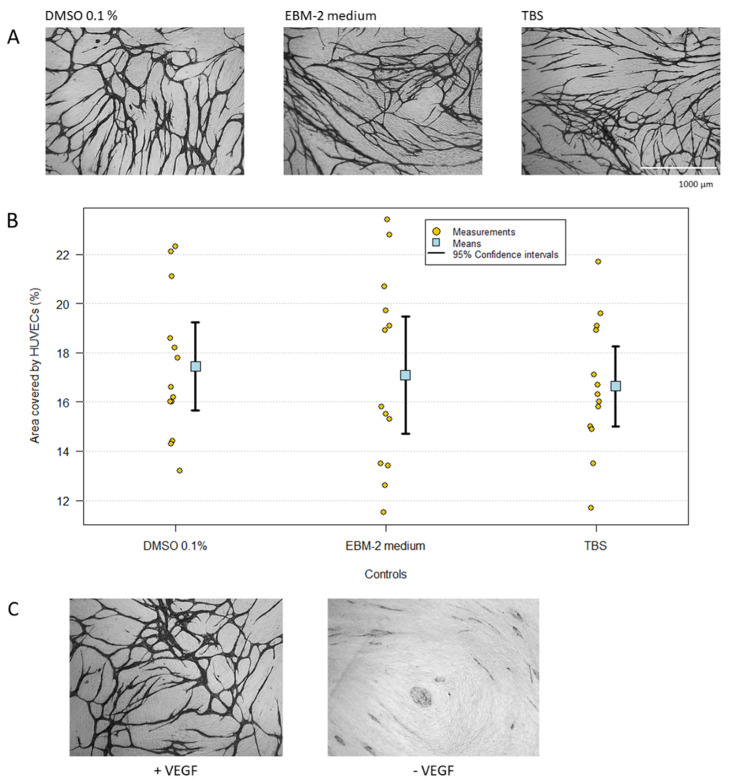
(**A**,**B**) Comparison of control groups. One-way ANOVA, F = 0.20 and P = 0.82. (**C**) The effect of VEGF on HUVEC differentiation. The HUVECs were cultivated in TFSM2-medium supplemented with and without VEGF. In presence of VEGF, the HUVECs form a capillary-like network. In the absence of VEGF, HUVECs form clusters and cords (magnification: 4×).

**Figure 2 ijms-23-04277-f002:**
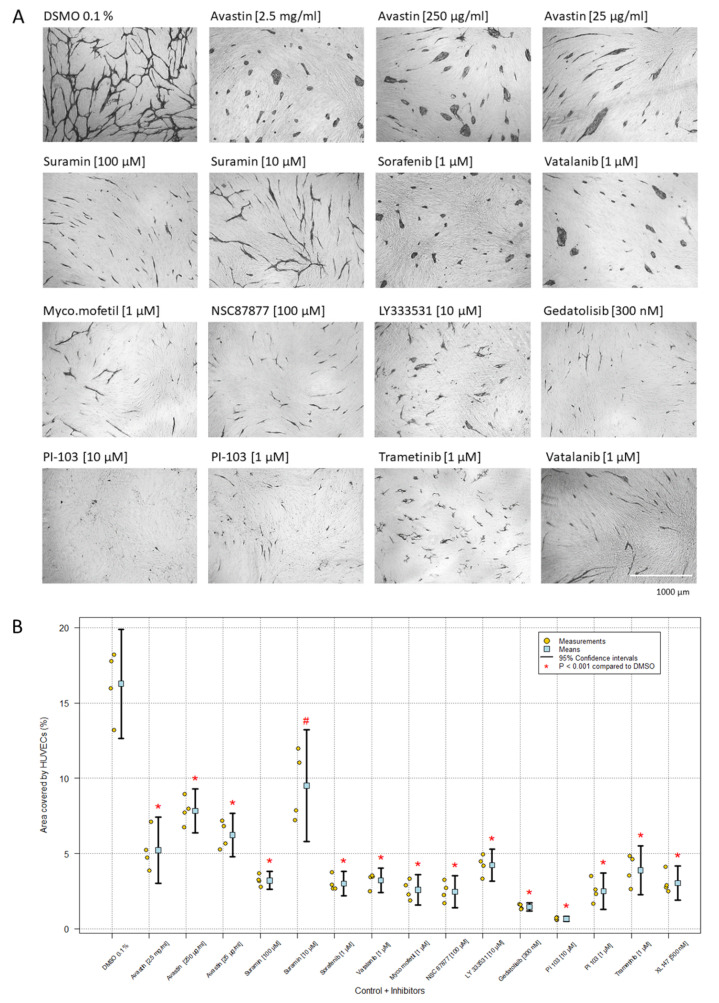
Representative images and image analysis results show the percentage of area covered by HUVECs after treatment with different inhibitors. (**A**) The morphology of the HUVECs was visualized by immunostaining with CD31 MAb. The images are based on one experiment representative of all experiments with these inhibitors (magnification: 4×). The concentrations of the inhibitors were chosen by the lowest concentration that gives a clear visual effect on the morphology without showing the general toxicity. (**B**) Data are illustrated in the plot as the mean of four independent experiments with duplicates and 95% confidence intervals. One-way ANOVA. F = 51.63 and P < 0.001. All treatments differed significantly from the control, except for suramin 10 µM (adjusted * *p*-values < 0.001, except # *p* = 0.028).

**Figure 3 ijms-23-04277-f003:**
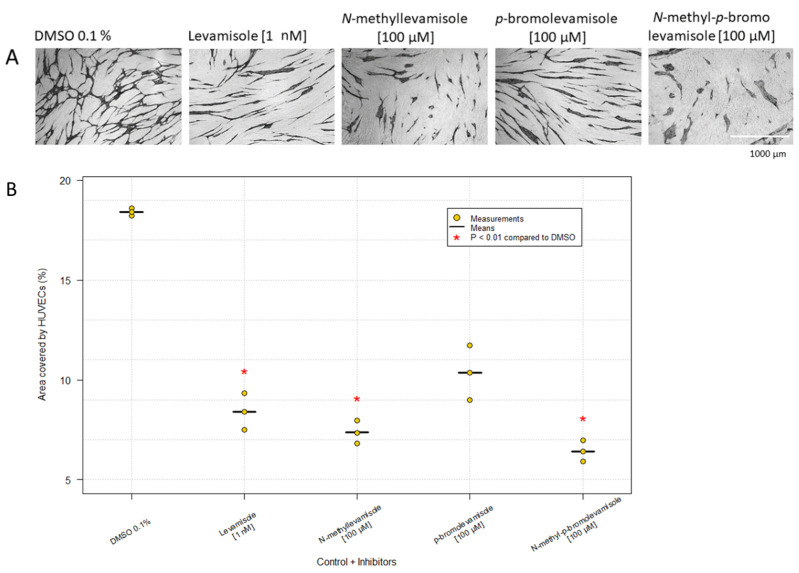
Effect of levamisole and derivatives. (**A**) HUVECs were treated with levamisole, *N*-methyllevamisole, *p*-bromolevamisole and *N*-methyl-*p*-bromolevamisole. HUVEC morphology was visualized by immunostaining with CD31 MAb. The images are based on one experiment, which is representative of three experiments, with duplicates (magnification: 4×). (**B**) Data are illustrated in the plot as the mean of three independent experiments with duplicates. Kruskal–Wallis *Χ*^2^ = 12.63. P = 0.013. *p*-bromolevamisole [100 µM] treatment did not differ from the control (adjusted *p*-value = 0.13).

**Figure 4 ijms-23-04277-f004:**
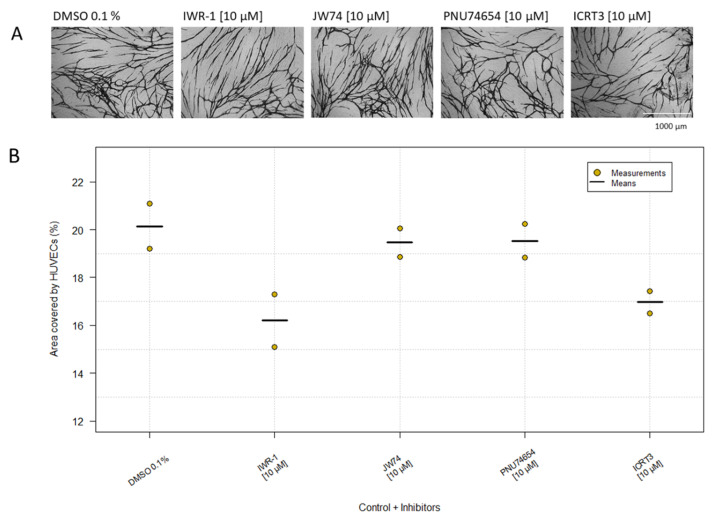
β-catenin inhibitors. DMSO 0.1% was used as control. Cells were treated with four different β-catenin inhibitors. (**A**) Morphology of the HUVECs was visualized by immunostaining with CD31 MAb in co-culture with NHDFs (magnification: 4×). (**B**) Data are shown in the plot as the mean of duplicates from two independent experiments compared with the control. Kruskal–Wallis *Χ*^2^ = 6.98. P = 0.14.

**Figure 5 ijms-23-04277-f005:**
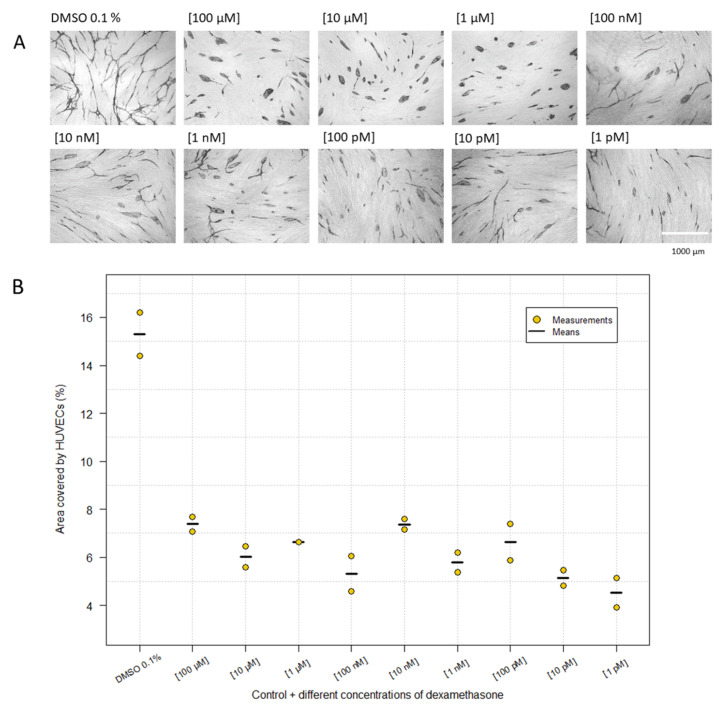
Effect of dexamethasone, in different concentrations, on HUVECs. The morphology (**A**) and area (**B**) of the HUVECs were visualized by immunostaining with CD31 MAb (magnification: 4×). (**B**) Image analysis shows the effect of dexamethasone on the percentage of area covered by HUVECs. Data are based on two independent experiments, which were carried out in duplicate. Kruskal–Wallis *Χ*^2^ = 16.23. P = 0.062.

**Figure 6 ijms-23-04277-f006:**
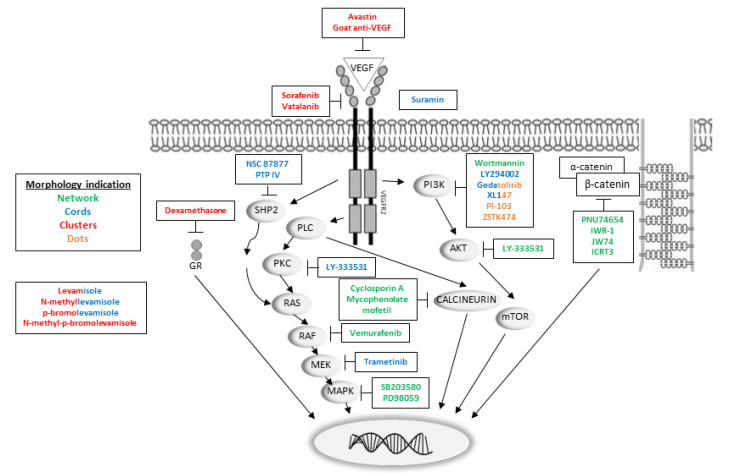
VEGFR2 downstream pathway with insertion of some of the inhibitor targets, with the HUVEC morphology indicated by colors. Green: network, blue: cords, red: clusters, orange: dots. Two colors indicate mixed morphology. The exact mechanism of action of suramin is still unknown, but it can bind to VEGF and VEGFR2’s co-receptor neuropilin, (not shown in the figure). From left to right: the SHP2-RAS-RAF-MEK-MAPK pathway, the PLC-PKC-RAS-RAF-MEK-MAPK pathway, the PLC-calcineurin pathway and the PI3K-AKT-mTOR pathway, which results in activation of key transcription factors and prepares the cell for differentiation and/or proliferation. Activation of PLC results in the hydrolysis of PIP2 and leads to generation of DAG and IP3. DAG is an activator of PKC, which among others activates the MAPK signaling pathway, including RAS, the serine/threonine protein kinase RAF and MEK. The RAS-RAF-MEK-MAPK pathway can also be activated by SHP2, which is coupled to VEGFR2 through Gab1. Another action of PLC is activation of calcineurin, a phosphatase that activates the transcription factor NFAT (nuclear factor of activated T cells). PI3K converts PIP2 to PIP3 by phosphorylation and PIP3 activates AKT, which can directly phosphorylate and activate mTOR, which regulates several TFs. Abbreviations: AKT: Ak strain transforming, PKB: protein kinase B, DAG: 1,2-diacylglycerol, Gab1: GRB2-associated-binding protein 1, GR: glucocorticoid receptor, IP3: inositol triphosphate, MAPK: mitogen-activated protein kinase, MEK: MAPK kinase, mTOR: mammalian target of rapamycin, PI3K: phosphatidylinositol 3-kinase, PIP2: phosphatidylinositol-4,5-bisphosphate, PIP3: phosphatidylinositol (3,4,5)-trisphosphate, PKC: protein kinase C, PLC: phospholipase C, RAS: rat sarcoma, RAF: rapidly accelerated fibrosarcoma, SHP2: Src homology phosphotyrosine phosphatase 2, VEGFR2: vascular endothelial growth factor receptor 2.

**Table 1 ijms-23-04277-t001:** Effect of inhibitors on HUVECs and GFP-HUVECs. Inhibitory rate is only given in relation to degree of network formation.

Target	Inhibitors	Concentration	HUVEC Morphology	Inhibitory Rate 0–5	GFP-HUVEC Morphology	Inhibitory Rate 0–5
Control	DMSO	0.1%	Network	0	Network	0
VEGF-A	Avastin	2.5 mg/mL	Clusters	-	NT	-
		250 µg/mL	Clusters + short cords	-	Clusters + short cords	-
		25 µg/mL	Clusters + short cords	-	Clusters + short cords	-
	Goat anti-VEGF	40 µg/mL	Clusters + short cords	-	NT	-
		5 µg/mL	Clusters +long cords	-	NT	-
VEGF/Neuropilin/PDGF/bFGF	Suramin	100 µM	Short cords	-	Short cords	-
		50 µM	Short cords	-	NT	-
		10 µM	Network	2	Network	2
VEGFR-2	Sorafenib	1 µM	Clusters	-	Clusters	-
		100 nM	Network	2	Network	-
		10 nM	Network	1	NT	-
		1 nM	Network	1	NT	-
	Vatalanib	10 µM	Clusters	-	Dead	-
		1 µM	Clusters	-	Clusters	-
		100 nM	Clusters + short cords	-	Clusters	-
		10 nM	Clusters + long cords	-	Clusters	-
β-catenin	IWR-1	10 µM	Network	0	Network	0
	PNU-74654	10 µM	Network	0	Network	0
	JW74	10 µM	Network	0	Network	0
	ICRT3	10 µM	Network	0	Network	0
Alkaline phosphatase	Levamisole	1 mM	Oval clusters + long cords	*-*	Oval clusters + network	*-*
	*P*-bromo-levamisole	1 mM	Dead	*-*	Dead	*-*
		100 µM	Network	2	Long cords	*-*
		10 µM	Network	0	Network	0
	*N*-methyl-levamisole	1 mM	Dead	*-*	Dots	*-*
		100 µM	Oval clusters + short cords	*-*	Clusters + short cords	*-*
		10 µM	Network	1	Long cords	*-*
	*N*-methyl- *p*-bromo-levamisole	1 mM	Dead	*-*	Dead	*-*
		100 µM	Oval clusters	-	Clusters	-
		10 µM	Network	2	Clusters + short cords	-
		1 µM	Network	0	Network	0
Calcineurin	Cyclosporin A	10 µM	Network	0	Dead	-
		1 µM	Network	0	Network	0
		100 nM	Network	0	Network	0
		10 nM	Network	0	Network	0
	Mycophenolate mofetil	10 µM	Short cords	-	Short cords	-
		1 µM	Short cords	-	Short cords	-
		100 nM	Network	1	Short cords	-
		10 nM	Network	0	Short cords	-
MAPK	SB203580	10 µM	Network	2	Network	1
		1 µM	Network	2	Network	1
		100 nM	Network	1	Network	1
		10 nM	Network	1	Network	1
	PD98059 ^1^	10 µM	Network	3	Long cords	-
		1 µM	Network	2	Network	2
		100 nM	Network	1	Network	1
		10 nM	Network	1	Network	1
MEK	Trametinib	1 µM	Curly short cords	-	Dead	-
		100 nM	Short cords	-	Dead	-
		10 nM	Short cords	-	Dead	-
		1 nM	Network	2	NT	-
RAF	Vemurafenib	10 µM	Dead	-	Dead	-
		1 µM	Network	0	Network	0
		100 nM	Network	0	Network	0
		10 nM	Network	0	Network	0
PI3K	Gedatolisib ^2^	30 µM	Dead	-	Dead	-
		300 nM	Short cords + dots	-	Dead	-
		30 nM	Network	1	Dots	-
		3 nM	Network	1	NT	-
	Wortmannin	10 µM	Network	3	Network	0
		1 µM	Network	2	Network	0
		100 nM	Network	2	Network	0
	ZSTK474	100 µM	Dead	-	NT	-
		10 µM	Dots	-	NT	-
		1 µM	Long cords	-	NT	-
		100 nM	Network	2	NT	-
		10 nM	Network	1	NT	-
		1 nM	Network	1	NT	-
	PI-103 ^2^	100 µM	Dead	-	NT	-
		10 µM	Dots	-	NT	-
		1 µM	Short cords + dots	-	NT	-
		100 nM	Long cords	-	NT	-
		10 nM	Network	2	NT	-
		1 nM	Network	1	NT	-
	XL147	50 µM	Dead	-	NT	-
		500 nM	Short cords + dots	-	NT	-
		50 nM	Network	0	NT	-
		5 nM	Network	0	NT	-
		500 pM	Network	0	NT	-
	LY294002	100 µM	Dead	-	NT	-
		10 µM	Short cords	-	NT	-
		1 µM	Network	1	NT	-
		100 nM	Network	0	NT	-
		10 nM	Network	0	NT	-
		1 nM	Network	0	NT	-
SHP1 and SHP2	PTP IV ^3^	100 µM	Dead	-	NT	-
		10 µM	Dead	-	NT	-
		1 µM	Long cords	-	NT	-
		100 nM	Network	1	NT	-
		10 nM	Network	0	NT	-
	NSC 87877 ^4^	100 µM	Short cords	-	NT	-
		10 µM	Network	1	NT	-
		1 µM	Network	0	NT	-
		100 nM	Network	0	NT	-
		10 nM	Network	0	NT	-
		1 nM	Network	0	NT	-
PKC	LY33353	50 µM	Dead	-	NT	-
		10 µM	Short cords	-	NT	-
		1 µM	Network	2	NT	-
		100 nM	Network	0	NT	-
		10 nM	Network	0	NT	-
		1 nM	Network	0	NT	-
AKT	Perifosine	50 nM	Network	2	Dead	-
		5 nM	Network	1	Network	0
		500 pM	Network	0	Network	0
STAT 5	STAT 5	100 µM	Short cords	-	NT	-
		10 µM	Network	1	NT	-
		1 µM	Network	0	NT	-
		100 nM	Network	0	NT	-
		10 nM	Network	0	NT	-
JAK-2	AG-490	100 µM	Dead	-	NT	-
		10 µM	Network	1	NT	-
		1 µM	Network	1	NT	-
		100 nM	Network	0	NT	-
		10 nM	Network	0	NT	-
JAK 1 & 2	Ruxolitinib	100 µM	Dead	-	NT	-
		10 µM	Network	1	NT	-
		1 µM	Network	0	NT	-
		100 nM	Network	0	NT	-
		10 nM	Network	0	NT	-
STAT 3	SH-4-54	100 µM	Dead	-	NT	-
		10 µM	Dead	-	NT	-
		1 µM	Network	1	NT	-
		100 nM	Network	0	NT	-
		10 nM	Network	0	NT	-
	NSC 74859	100 µM	Dead	-	Dots	-
		10 µM	Network	0	Network	0
		1 µM	Network	0	Network	0
JAK 2 & STAT 3	WP1066	100 µM	Dead	-	NT	-
		10 µM	Dead	-	NT	-
		1 µM	Network	0	NT	-
		100 nM	Network	0	NT	-
		10 nM	Network	0	NT	-
GR	Dexamethasone	1 mM	Dead	-	NT	-
		100 µM	Cluster	-	NT	-
		10 µM	Cluster	-	NT	-
		1 µM	Cluster	-	NT	-
		100 nM	Cluster + cords	-	NT	-
		10 nM	Cluster + cords	-	NT	-
		1 nM	Cluster + cords	-	NT	-
		100 pM	Cluster + cords	-	NT	-
		10 pM	Cluster + cords	-	NT	-
		1 pM	Cluster + cords	-	NT	-
PP1	Tautomycetin	1 µM	Network	0	NT	-
		100 nM	Network	0	NT	-
		10 nM	Network	0	NT	-
		1 nM	Network	0	NT	-
		100 pM	Network	0	NT	-
Histone deacetylase and histone acetyl-transferase	Butyric acid	1 mM	Short cords	-	NT	-
		100 µM	Network	0	NT	-
		10 µM	Network	0	NT	-
		1 µM	Network	0	NT	-
		100 nM	Network	0	NT	-
		10 nM	Network	0	NT	-
	Valproic acid	1 mM	Short cords	-	NT	-
		100 µM	Network	1	NT	-
		10 µM	Network	0	NT	-
		1 µM	Network	0	NT	-
		100 nM	Network	0	NT	-
		10 nM	Network	0	NT	-
	Anacardic acid	100 µM	Dead	-	NT	-
		10 µM	Dead	-	NT	-
		1 µM	Network	0	NT	-
		100 nM	Network	0	NT	-
		10 nM	Network	0	NT	-
		1 nM	Network	0	NT	-
		100 pM	Network	0	NT	-
Retinoic acidreceptor (RAR)	Retinol	100 µM	Dead	-	NT	-
		10 µM	Network	0	NT	-
		1 µM	Network	0	NT	-
		100 nM	Network	0	NT	-
		10 nM	Network	0	NT	-
		1 nM	Network	0	NT	-
	Retinoic acid	100 µM	Dead	-	NT	-
		10 µM	Network	0	NT	-
		1 µM	Network	0	NT	-
		100 nM	Network	0	NT	-
		10 nM	Network	0	NT	-
VDR	25-hydroxyvitamin D3 monohydrate	1 µM	Short cords	-	NT	-
		100 nM	Network	0	NT	-
		10 nM	Network	0	NT	-
		1 nM	Network	0	NT	-
		100 pM	Network	0	NT	-
	25-hydroxyvitamin D3 monohydrate + vitamin D-binding protein (DBP)	1 µM	Short cords	-	NT	-
		100 nM	Network	0	NT	-
		10 nM	Network	0	NT	-
		1 nM	Network	0	NT	-
		100 pM	Network	0	NT	-

^1^ PD98059 is a potent and selective inhibitor of MAP kinase kinases (MAPKK), MEK1 and MEK2 [1]. It binds to the inactive form of MAPKK and prevents activation by upstream activators such as c-Raf. ^2^ Gedatolisib and PI-103 are potent PI3K/mTOR inhibitors, inhibiting both PI3K and mTOR kinases; they are more specific to PI3K. ^3^ PTP IV is reported to inhibit SHP-2 (IC_50_ = 1.8 µM). ^4^ NSC 87877 is a cell-permeable inhibitor of both SHP-1 and SHP-2 (IC_50_ = 355 and 318 nM, respectively).

**Table 2 ijms-23-04277-t002:** Effect of combinations of inhibitors on HUVECs. Inhibitory rate is only given in relation to degree of network formation.

Combinations of Inhibitors	HUVEC Morphology	Inhibitory Rate 0–5
LY333531 [10 µM] NSC 87877 [100 µM] Gedatolisib [300 nM]	Dots	*-*
LY294002 [10 µM] LY333531 [10 µM] NSC 87877 [100 µM]	Cords	*-*
Perifosine [50 nM] LY333531 [10 µM] NSC 87877 [100 µM]	Network	3
NSC 87877 [100 µM] LY294002 [10 µM]	Cords	*-*
NSC 87877 [100 µM] GEDA [30 nM]	Network	3
LY333531 [10 µM] PTP IV [1 µM]	Cords	*-*
LY294002 [10 µM] LY333531 [10 µM]	Cords	*-*
LY333531 [10 µM] NSC 87877 [100 µM]	Dots	*-*
PTP IV [100 nM] GEDA [30 nM]	Network	3
Perifosine [50 nM] LY333531 [10 µM] NSC 87877 [100 µM]	Network	3
Perifosine [50 nM] LY333531 [10 µM]	Network	2
Suramin [100 µM] AG-490 [10 µM]	Network	3
Sorafenib [1 µM] STAT 5 [100 µM]	Network	4
Suramin [100 µM] STAT 5 [100 µM]	Network	4
Sorafenib [1 µM] NSC 74859 [100 µM]	Cords	*-*
Avastin [250 µg/mL] STAT 5 [100 µM]	Cords	*-*
Suramin [100 µM] NSC 74859 [100 µM]	Cords	*-*
PTP IV [1 µM] NSC 87877 [100 µM]	Cords	*-*
Trametinib [100 nM] LY333531 [10 µM]	Cords	*-*
Sorafenib [1 µM] Vatalanib [1 µM]	Clusters	*-*
Avastin [25 µg/mL] Vatalanib [1 µM]	Clusters	*-*
Poly goat antibody [5 µg/mL] Vatalanib [1 µM]	Clusters	*-*
Vatalanib [1 µM] AG-490 [10 µM]	Clusters	*-*
Vatalanib [1 µM] NSC 74859 [100 µM]	Clusters	*-*
Avastin [250 µg/mL] NSC 74859 [100 µM]	Clusters	*-*
Vatalanib [1 µM] STAT 5 [100 µM]	Clusters	*-*
Avastin [250 µg/mL] AG-490 [10 µM]	Clusters + cords	*-*
NSC 87877 [100 µM] Vatalanib [1 µM]	Clusters + dots	*-*
Myco. Mofetil [10 µM] Sorafenib [1 µM]	Dead	*-*
ZSTK474 [10 µM] PI-103 [10 µM]	Dead	*-*
LY333531 [10 µM] PI-103 [10 µM]	Dead	*-*
LY333531 [10 µM] PI-103 [10 µM] Sorafenib [10 µM] PTP IV [10 µM]	Dead	*-*
LY333531 [10 µM] PI-103 [10 µM] Sorafenib [1 µM] NSC 87877 [100 µM]	Dead	*-*

**Table 3 ijms-23-04277-t003:** Grading of results based on the percentage of area covered by HUVECs. To evaluate the effect of the inhibitors, both the percentage of area covered by HUVECs and their morphology were recorded. Cells that form clusters can still yield high coverage as proliferation is not affected, showing the importance of comparing the quantitative image analysis results with the qualitative results, nor is the morphology.

HUVEC Coverage Area %	HUVEC Coverage Area Grading	Inhibitory Rate 0–5
0–3%	+	5
3–6%	++	4
6–9%	+++	3
9–12%	++++	2
12–18%	+++++	1
>18%	++++++	0

## Data Availability

The data that support the findings of this study are available from the corresponding author on reasonable request.

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
