# Peer review of "In Vitro Angiogenesis Inhibition and Endothelial Cell Growth and Morphology"

_ijms, 2022, doi:10.3390/ijms23084277_

Round 1

Reviewer 1 Report

The paper describes a co-culture angiogenesis assay with HUVECs and dermal fibroblasts used to investigate the effect of several angiogenesis inhibitors, in order to evaluate effective cancer therapies. The authors declare that the advantage of this assay is that it allows both a qualitative and a quantitative evaluation of the effect of different compounds.      
The study could be interesting for the audience of the journal but different points should be extensively improved and require major revision:

  1. In table 1, the degree of inhibition of network formation of each tested compound on HUVEC is summarized, (the author said that 0: no inhibition, 5: 100 % inhibition) but is not clear how is calculated this degree. Please explain the method of quantification and detail the different degrees.
  2. The quality of the figures is very poor, please increase the resolution and/or the dimension of the network images and decrease the histograms ones (please note that in figure 7 you used a different histogram, please uniform them)
  3. The statistical analysis of the data is lacking and this is demonstrated by the absence of any statistical significance (*) reported on the graphs. The authors say: “the images are based on one experiment, which is representative for two experiments (n=2 with duplicates)”, “the experiment was made twice with duplicate” or “data is based on one experiment ± SD, which was made in duplicates”. It’s impossible to run one way anova, as declared by the authors in the methods, with such few amount of data. At least three experiments in triplicates are necessary to run a solid statistical analysis. Please show them and the relative p-values. If no statistical significances arise from the new analysis, please revise your conclusion.

Minor:

Figure 1 and figure 8 are almost identical and may be merged in one single figure. I suggest to maintain only figure 8.

Author Response

Responses to reviewers

Reviewer 1

The paper describes a co-culture angiogenesis assay with HUVECs and dermal fibroblasts used to investigate the effect of several angiogenesis inhibitors, in order to evaluate effective cancer therapies. The authors declare that the advantage of this assay is that it allows both a qualitative and a quantitative evaluation of the effect of different compounds.      
The study could be interesting for the audience of the journal but different points should be extensively improved and require major revision:

  1. In table 1, the degree of inhibition of network formation of each tested compound on HUVEC is summarized, (the author said that 0: no inhibition, 5: 100 % inhibition) but is not clear how is calculated this degree. Please explain the method of quantification and detail the different degrees.

The method of quantification has been explained and described in detail (new Table 3 in materials and methods). We want to emphasize that the effects of different inhibitors must be evaluated by a combination of morphology and area coverage, as described in the manuscript.

  1. The quality of the figures is very poor, please increase the resolution and/or the dimension of the network images and decrease the histograms ones (please note that in figure 7 you used a different histogram, please uniform them).

We have decreased the no of pictures and increased the quality of the pictures as requested.

  1. The statistical analysis of the data is lacking and this is demonstrated by the absence of any statistical significance (*) reported on the graphs. The authors say: “the images are based on one experiment, which is representative for two experiments (n=2 with duplicates)”, “the experiment was made twice with duplicate” or “data is based on one experiment ± SD, which was made in duplicates”. It’s impossible to run one way anova, as declared by the authors in the methods, with such few amount of data. At least three experiments in triplicates are necessary to run a solid statistical analysis. Please show them and the relative p-values. If no statistical significances arise from the new analysis, please revise your conclusion.

We have carried out statistical analysis by ANOVA and Kruskal-Wallis test as appropriate and indicated p values.

Minor:

Figure 1 and figure 8 are almost identical and may be merged in one single figure. I suggest to maintain only figure 8.

Amended as requested.

Reviewer 2 Report

Submitted for review
the article contains the results of inhibition of the formation of vessel-like structures from HUVEC using inhibitors with different mechanisms of action. Among the main shortcomings, it should be noted a simplified method for assessing the morphology of HUVEC, the formation of a vessel-like network, the lack of justification for the choice of this set of inhibitors, as well as the excessive description of some sections.
Notes:
1. The authors do not represent the methodology by which the scoring was carried out
 inhibitory rate.
2. The authors do not give reasons why they did not use the inhibition assessment based on a more accurate determination of the quantitative parameters of the network: the area, the length of the structures, the number of branching points, and similar criteria that can be quite easily obtained by processing the data.
3. Data on the cytotoxic effect of these drugs on HUVEC and GFP-HUVEC are not provided. Therefore, it can be assumed that many of the observed effects are the result of a cytotoxic effect, and not inhibition of the VEGF-VEGFRs interaction.
4. The authors give an oversimplified discussion of the results obtained.
5. For substances with uncertain effects on angiogenesis processes (Levamisole and its derivatives), inhibition of VEGFR2-dependent signaling pathways should be shown by standard methods, for example, using Western blotting.

Author Response

Responses to reviewers

Reviewer 2

Submitted for review
the article contains the results of inhibition of the formation of vessel-like structures from HUVEC using inhibitors with different mechanisms of action. Among the main shortcomings, it should be noted a simplified method for assessing the morphology of HUVEC, the formation of a vessel-like network, the lack of justification for the choice of this set of inhibitors, as well as the excessive description of some sections.

The choice of inhibitors was guided by the wish to evaluate the contribution of major VEGFR signalling pathways. Not only by their quantitative effect but also by their effect on morphology. This we believe to be a major strength of the assay used and an aspect not described much in the literature. Moreover, we selected additional potential inhibitors by their availability and possible interest.

Notes:
1. The authors do not represent the methodology by which the scoring was carried out
 inhibitory rate.

The method of quantification has been explained and described in detail (Table 3 in materials and methods)

  1. The authors do not give reasons why they did not use the inhibition assessment based on a more accurate determination of the quantitative parameters of the network: the area, the length of the structures, the number of branching points, and similar criteria that can be quite easily obtained by processing the data.

The effects of the inhibitors were evaluated by their quantitative effect and by their effect on morphology, which we believe to be a major strength of the assay used and an aspect not described much in the literature. In the quantitative evaluation, we used area %

In the qualitative evaluation, we used morphology, which clearly reflect the effect on the VEGFR as demonstrated in Fig. 1 and Fig. 2 by omission of VEGF from the culture medium and by addition of VEGF.neutralizing antibodies.

  1. Data on the cytotoxic effect of these drugs on HUVEC and GFP-HUVEC are not provided. Therefore, it can be assumed that many of the observed effects are the result of a cytotoxic effect, and not inhibition of the VEGF-VEGFRs interaction.

The general cytotoxic effect was evaluated by the integrity of the NHDF layer, on which the HUVECs were grown (visual integrity as seen by microscopy). Only concentrations, not affecting the NHDF were used. Moreover, the presence and integrityof the HUVECs was evaluated by visual inspection (microscopy) and by staining with a Mab to CD31.

  1. The authors give an oversimplified discussion of the results obtained.

The discussion has been improved according to the comments of reviewer 1 and 2.

  1. For substances with uncertain effects on angiogenesis processes (Levamisole and its derivatives), inhibition of VEGFR2-dependent signaling pathways should be shown by standard methods, for example, using Western blotting.

We have amended the discussion and conclusion on this part to merely state that levamisole and its derivatives may affect the VEGFR indirectly or directly and that this should be investigated in future experiments. At present, we are regrettably not in a position to carry out these experiments.

Round 2

Reviewer 1 Report

The authors answered to all the queries arisen by the reviewers and now the article is clear, consistent and worthy. I suggest the acceptance of their paper in the journal.

Reviewer 2 Report

Thank you for your replies and the changes made to the article.